

# Automated, phylogeny-based genotype delimitation of the Hepatitis Viruses HBV and HCV

Dora Serdari[1], Evangelia-Georgia Kostaki[2], Dimitrios Paraskevis[2], Alexandros Stamatakis[1,3] and Paschalia Kapli[4]

[1] The Exelixis Lab, Scientific Computing Group, Heidelberg Institute for Theoretical Studies, Heidelberg, Germany

[2] Department of Hygiene Epidemiology and Medical Statistics, School of Medicine, National and Kapodistrian University of Athens, Athens, Greece

[3] Institute for Theoretical Informatics, Karlsruhe Institute of Technology, Karlsruhe, Germany

[4] Centre for Life's Origins and Evolution, Department of Genetics Evolution and Environment, University College London, University of London, London, United Kingdom

Corresponding author
Paschalia Kapli, p.kapli@ucl.ac.uk, k.pashalia@gmail.com

## ABSTRACT

**Background**. The classification of hepatitis viruses still predominantly relies on ad hoc criteria, i.e., phenotypic traits and arbitrary genetic distance thresholds. Given the subjectivity of such practices coupled with the constant sequencing of samples and discovery of new strains, this manual approach to virus classification becomes cumbersome and impossible to generalize.

**Methods**. Using two well-studied hepatitis virus datasets, HBV and HCV, we assess if computational methods for molecular species delimitation that are typically applied to barcoding biodiversity studies can also be successfully deployed for hepatitis virus classification. For comparison, we also used ABGD, a tool that in contrast to other distance methods attempts to automatically identify the barcoding gap using pairwise genetic distances for a set of aligned input sequences.

**Results—Discussion**. We found that the mPTP species delimitation tool identified even without adapting its default parameters taxonomic clusters that either correspond to the currently acknowledged genotypes or to known subdivision of genotypes (subtypes or subgenotypes). In the cases where the delimited cluster corresponded to subtype or subgenotype, there were previous concerns that their status may be underestimated. The clusters obtained from the ABGD analysis differed depending on the parameters used. However, under certain values the results were very similar to the taxonomy and mPTP which indicates the usefulness of distance based methods in virus taxonomy under appropriate parameter settings. The overlap of predicted clusters with taxonomically acknowledged genotypes implies that virus classification can be successfully automated.

# INTRODUCTION

The continuous advances in next generation sequencing technologies lead to an increasingly easier and inexpensive production of genome and metabarcoding data. The wealth of

available data has triggered the development of new models of molecular evolution, algorithms, and software, that aim to improve molecular sequence analyses in terms of biological realism, computational efficiency, or a trade-off between the two. In response to such technological and technical advancements, several fields of biology have undergone a substantial transformation. Sequence-based species delimitation and identification, in the framework of DNA-(meta)barcoding constitutes a representative example that revived taxonomy and systematics (*Tautz et al., 2003*; *Moritz & Cicero, 2004*; *Savolainen et al., 2005*; *Waugh, 2007*; *Bucklin et al., 2010*; *Valentini, Pompanon & Taberlet, 2009*; *Li et al., 2015*), while it also provided a new means of analysis in several fields (*Galimberti et al., 2013*; *Mishra et al., 2016*; *Leray & Knowlton, 2015*; *Bell et al., 2016*; *Batovska et al., 2018*). Among others, the development of novel species delimitation tools has substantially advanced the study of biodiversity of microorganism that are often hard to isolate and study (*Taberlet et al., 2012*; *Gibson et al., 2014*; *Thomsen & Willerslev, 2015*). The sequencing of environmental samples in conjunction with algorithms for genetic clustering has led to the identification of a plethora of previously unknown organisms and a re-assessment of the microbial biodiversity in several settings.

In a similar context, genetic information has been a rich source of information for viral species. Several studies show how phylogenetic information can be deployed for identifying the spatial and temporal origin of a virus, potential factors that trigger its dispersal, and other key epidemiological parameters (*Stadler et al., 2011*; *Stadler et al., 2014*; *Gire et al., 2014*). In an era of high human mobility, such methods are important, as the increase of emerging and re-emerging epidemics is even more prominent than in the past (*Balcan et al., 2009*; *Meloni et al., 2011*; *Pybus, Tatem & Lemey, 2015*). Nevertheless, phylogenetic information is still not used in the context of virus species classification or identification. As we have witnessed for other microorganisms, using or adapting already available methods for fast and automated delimitation or identification of virus species can greatly contribute to better understand their evolution.

To date, the official taxonomy of viruses (ICTV, i.e., International Committee on Taxonomy of Viruses) has mainly been based on established biological classification criteria as used for other life forms, such as plants or animals. An analogous hierarchical classification system containing orders, families, subfamilies, genera, and species is being applied (*Simmonds, 2015*). The ICTV is typically based on phenotypic criteria, such as morphology, nucleic acid type (i.e., DNA or RNA), hosts, symptoms, mode of replication, geographical data, or presence of antigenic epitopes, to name a few. Generally, such criteria, despite being informative, can be subjective, require highly specialized knowledge, and are time consuming to apply. In contrast, sequence evolution takes into account the evolutionary history of life forms and, thus, may offer a more objective source of information for taxonomic classification. An important difference in viruses compared to other organisms is that they lack a common set of universal genes such as the 18S rRNA in eukaryotes or the 16S rRNA in prokaryotes. Therefore, we cannot infer a comprehensive virus tree of life (*Simmonds et al., 2017*), and, more importantly for species delimitation, we cannot rely upon barcoding markers that are universally suitable for all viruses. We can nonetheless gain valuable insights for their systematics by

utilizing phylogenetic information at lower taxonomic ranks (e.g., families, genera, species), using appropriate genes for each dataset. In this context, methods using genetic-distance thresholds (*Bao, Chetvernin & Tatusova, 2014*; *Lauber & Gorbalenya, 2012*; *Yu et al., 2013*) have been suggested as a complementary method to the traditional virus classification for accelerating new species identification.

In this study, we explore whether a recently developed algorithm for molecular species delimitation on barcoding or marker gene phylogenies can be deployed for ICTV. In contrast to genetic distance-based methods the multi-rate Poisson Tree Processes (mPTP, *Kapli et al., 2017*) infers the number of genetic clusters given a phylogenetic input tree. Such trees can easily be inferred using both, Maximum Likelihood (*Stamatakis, 2014*), or Bayesian approaches (*Ronquist et al., 2012*) on single-gene or multi-gene multiple sequence alignments. The fundamental assumption of the model is that variance in the data, as represented by the phylogeny, is greater among species than within a species (*Zhang et al., 2013*). The additional assumption of mPTP, that the genetic variation may differ substantially among species allows to accurately delimit species in large (meta-) barcoding datasets comprising multiple species of diverse life histories (*Kapli et al., 2017*). Experiments using empirical data for several animal phyla (*Kapli et al., 2017*) and recently also viruses (*Thézé et al., 2018*; *Modha et al., 2018*) show that the method consistently provides extremely fast and sensible species estimates on 'classic' phylogenetic marker and barcoding genes.

To assess whether mPTP can be deployed as a quantitative ICTV method we analyze two medically important viruses, Hepatitis B (HBV) and Hepatitis C (HCV), that are leading global causes of human mortality (*Stanaway et al., 2016*). Both viruses cause liver inflammation, but are substantially different from each other. HBV has a partially double-stranded circular DNA genome with a length of about 3.2 kb while HCV is a single-stranded, positive-sense RNA virus, with a genome length of approximately 10 kb (*Radziwill, Tucker & Schaller, 1990*; *Tang & McLachlan, 2001*; *Martell et al., 1992*). Both virus types comprise at least two taxonomic levels (HBV: genotypes, subgenotypes; HCV: genotypes, subtypes). Besides the significance of the two viruses for human health, we selected them as test cases since due to the substantial amount of taxonomic research that has been conducted and that we can hence use to assess the efficiency of genetic clustering (e.g., *Simmonds et al., 2005*; *Schaefer, 2007*; *Smith et al., 2014*; *Messina et al., 2015*).

## MATERIAL AND METHODS

### Datasets

We generated multiple sequence alignments (MSAs) corresponding to two virus types: HBV and HCV from two sets of full-length genomic sequences downloaded from publicly available databases (NCBI: http://www.ncbi.nlm.nih.gov/, accession numbers provided in Appendix S1).

The HBV dataset comprises 110 sequences corresponding to eight genotypes (i.e., A-H) and 31 subgenotypes. The genotypes (A through D, F, and H) have been further divided into subgenotypes indexed by numbers for the corresponding genotype (e.g., $A_1$, $A_2$, $B_1$, $B_2$,

B₃, etc.; *Kramvis, 2014*). The inter-genotypic and inter-subgenotypic divergence exceeds 8% and 4%–8%, respectively across the genome. No subgenotypes have been reported for genotypes E, G and H which shows that they are of lower levels of genetic divergence than the rest. The distribution of HBV genotypes differs greatly with respect to the geographical origin. Moreover, they differ in their natural history, response to treatment and disease progression (*Huang et al., 2013*; *Biswas et al., 2013*; *Moura et al., 2013*; *Shi et al., 2013*). For our study we included the sequences of the eight genotypes (A–H) that form part of the oldest identified HBV groups.

The HCV dataset (I) comprises 213 sequences corresponding to seven major taxonomic units named after genotypes (1, 2, 3, 4, 5, 6, and 7) and numerous subtypes (*Smith et al., 2014*). The HCV classification into genotypes and subtypes was based on genetic-distance thresholds that were verified by the fact that they formed monophyletic clades in an inferred phylogeny (*Smith et al., 2014*). Therefore, the HCV classification serves as an appropriate test case for assessing whether a similar clustering can be identified with a more objective and automated method, such as mPTP, that does not require any user input apart from a phylogeny.

## Genetic cluster delimitation

To delimit the putative species, additionally to mPTP, we used the distance-based "*Automatic Barcode Gap Discovery*" tool (ABGD, *Puillandre et al., 2012*). ABGD is a popular distance-based barcoding method that, compared to other distance-based methods attempts to automatically identify the threshold value for the transition from intra-specific variation to inter-specific divergence (*Puillandre et al., 2012*).

For the mPTP delimitation, a fully binary (bifurcating) rooted phylogeny is required. Therefore, using the aligned sequences we inferred the phylogenetic relationships under the GTR+Γ model of nucleotide substitution using RAxML-NG (*Kozlov et al., 2018*). We rooted the phylogenetic trees according to the originally published phylogenies (i.e., using the branch leading to genotypes F/H for HBV and genotype 7 for HCV). Using heuristic search algorithms for finding the 'best' delimitation given the rooted phylogeny and without any further prior assumptions. We performed the mPTP delimitation under Maximum Likelihood (ML) and calculated the support of the delimited clusters using Markov-chain Monte Carlo (MCMC) sampling (*Kapli et al., 2017*). We conducted the MCMC sampling twice for $10^6$ generations, to identify potential lack of convergence with a sampling frequency of 0.1.

For ABGD, the user has to define two important parameters, (i) the prior maximum divergence of intraspecific diversity ($P$), which implies that the barcode gap is expected to exceed this value and should not be confused with the genetic thresholds assumed to define the inter-specific relationships, (ii) a proxy for the minimum gap width ($X$), which indicates that the barcoding gap is expected to be $X$ times larger than any intraspecific gap (*Puillandre et al., 2012*). For both, HBV, and HCV, we used 10 prior maximum thresholds in the range of $p = 0.001$ and $P = 0.05$. The proxy for the minimum gap width ($X$) was set to the default value ($X = 1.5$) for HCV, while for HBV the default value did not yield any delimitation and we therefore set it to a lower value ($X = 0.5$).

## RESULTS & DISCUSSION

The biodiversity of viruses is tremendous and it is broadly accepted that our understanding of their ecology and evolution is constrained to a small fraction of species (*Paez-Espino et al., 2016*). In just a kilo of marine sediment there can be a million of different viral genotypes (*Breitbart & Rohwer, 2005*), while on a global scale the number of viruses is 10 million-fold higher than the number of stars in the universe (*Suttle, 2013*). The classification of such a diverse set of organisms constitutes a challenging task and is impossible to accomplish within reasonable time using phenotypic characters. Quantitative computational methods could provide a viable alternative, particularly for large scale clustering and fast identification of viral strains (*Simmonds et al., 2017*; *Modha et al., 2018*). Using empirical data of the HBV and HCV viruses we show that by applying phylogeny-aware and distance-based tools to classify the strains of the two virus types, the corresponding genetic clustering closely recovers their currently accepted taxonomy.

### HCV Clustering

The current taxonomy of HCV comprises seven genotypes, while mPTP yielded 16 genetic clusters (Fig. 1, Figs. S1 and S2 and Appendix S1). From the 16 clusters, five were congruent with the current taxonomy, i.e, genotypes 1, 2, 4, 5 and 7. On the contrary, genotype 3 and genotype 6 were further split into three and eight sub-clusters correspondingly (Fig. 1), which corroborates former views that divergent variants of these genotypes may qualify as separate major genotypes (*Simmonds et al., 2005*; *Smith et al., 2014*). In particular, the additional clusters identified by mPTP correspond to previously identified groups of subtypes (Fig. S1). For genotype 6, these clusters consisted of the following subtype groups: 6a; 6b and 6xd; 6c, 6d, 6e, 6f, 6g, 6o, 6p, 6q, 6r, 6s, 6t, 6u, 6w, 6xc and 6xf; 6 h, 6i, 6j, 6k, 6l, 6m, 6n, 6xb, 6xe; 6xa; 6v (Fig. S1 and Appendix S1). Similarly, for genotype 3, the delimited clusters were (i) 3g, 3b, 3i, 3a, 3e, 3d, (ii) 3k, and (iii) 3 h and 3. All clusters were substantially supported by the MCMC sampling, except the split of 3k subtype from its sister group (Fig. 1), which may be due to the limited amount of corresponding sequences.

The number of clusters inferred with ABGD ranged from 1 to 208 depending on the value of the maximum intraspecific divergence threshold (Fig. 2). The most reasonable result (i.e., the one closest to the current standard taxonomy) comprised 19 clusters and was obtained for a minimum of intraspecific genetic diversity of 5.99% (i.e., $p = 0.0599$). Under this threshold, the delimitation is largely identical to the delimitation obtained with mPTP (Fig. 1), with three differences: (i) that genotype 3 was split into four clusters, instead of three, (ii) genotype six was divided into nine clusters instead of eight, and, (iii) genotype 7 is divided into two clusters. When the prior intraspecific divergence was increased to a higher minimum of 10%, all sequences were grouped in a single cluster. When the threshold was set to a lower value (3.6%) the number of clusters increased to 135 (Fig. 2). Nevertheless, the delimitation with the 5.99% threshold is largely congruent to current taxonomy and the clusters obtained with mPTP, thus indicating the usefulness of distance-based methods in virus taxonomy under well informed parameters.

So far, the classification of HCV into genotypes and subtypes has been defined mostly by visual identification of clades in phylogenetic inference of HCV sequences (*Simmonds*

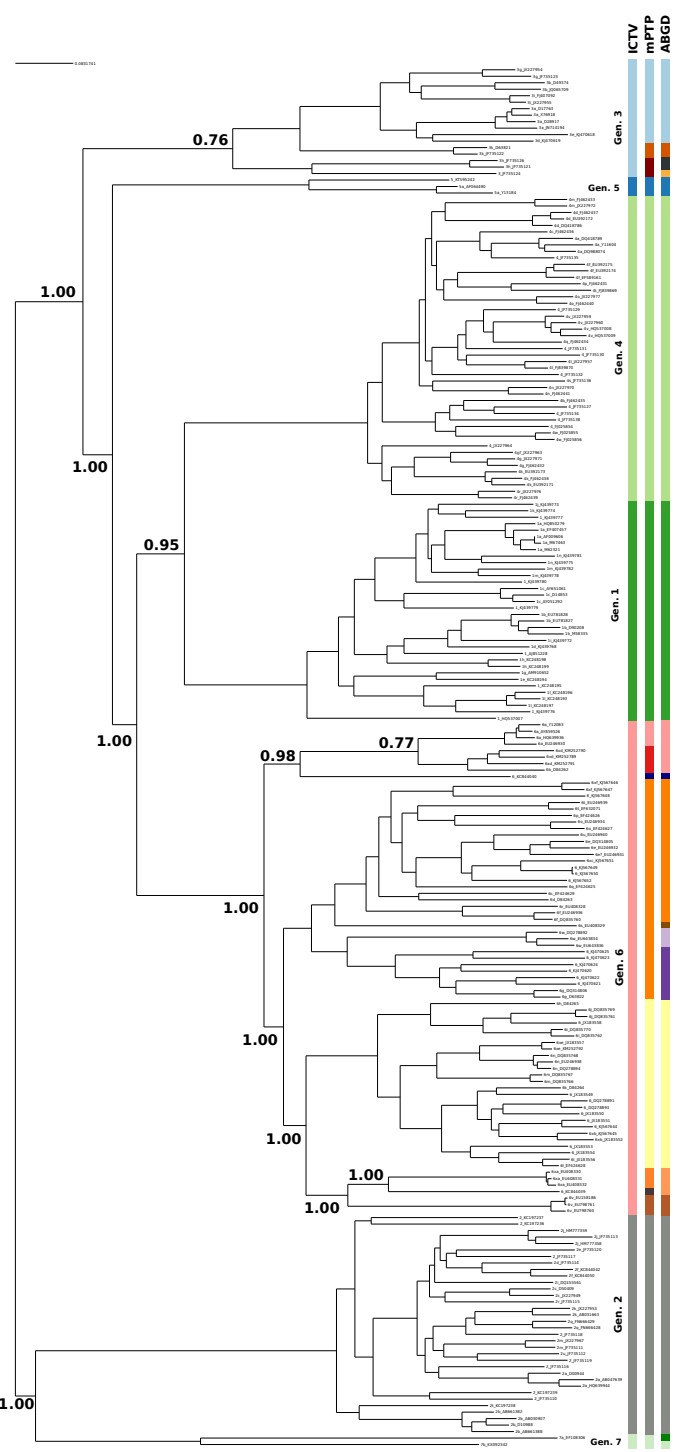

**Figure 1** Clustering of the HCV samples into genotypes; the first bar of colors corresponds to the genotypes currently acknowledged by ICTV, the second to the mPTP ML clustering and the third to the ABGD clustering ($p = 0.0599, X = 1.5$). The numbers indicate the support for a particular node being a speciation node obtained by the MCMC sampling under the mPTP model (support $<0.5$ not shown, but see Fig. S2). The phylogenetic relationships were inferred using RAxML under the GTR+$\Gamma$ model.

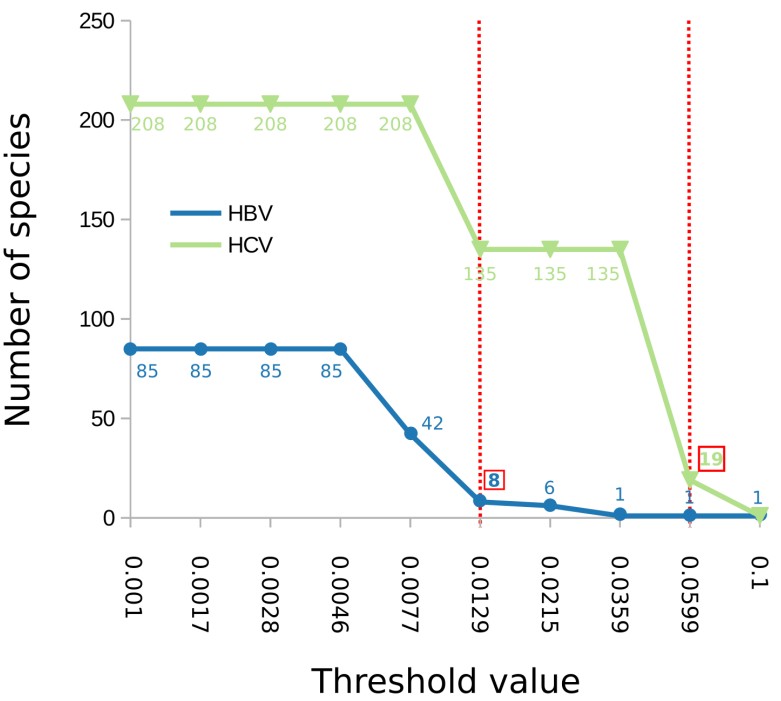

**Figure 2  Number of delimited clusters for ABGD with respect to input parameters.** The graph shows the change of the number of delimited clusters ($y$ axis) with respect to the minimum intraspecific threshold ("p") assumed by ABGD ($x$ axis). The threshold that yielded the most sensible clustering for HBV was $p = 0.0129$ while for HCV was $p = 0.0599$, both are shown with a dotted red line in the figure; the corresponding number of clusters is indicated in a red box.

*et al., 2005*; *Smith et al., 2014*). Specifically, the genotypes correspond to the seven major highly-supported phylogenetic HCV clusters while subtypes were defined as the secondary hierarchical clusters found within each genotype (*Smith et al., 2014*). This classification scheme has been widely adopted (*Combet et al., 2007*; *Yusim et al., 2016*) and has been shown to be robust (in terms of stability of the HCV phylogeny) and relevant for clinical practice, since response rates to immunomodulatory treatment for the chronic hepatitis C differs across genotypes. Nevertheless, new, unassigned lineages are often discovered from understudied areas (*Sulbaran et al., 2010*; *Nakano et al., 2012*; *Lu et al., 2013*; *Tong et al., 2015*) and it is challenging to assign them a taxonomic status, given that the genetic distance cut-off among intra and inter-specific relationships is arbitrary and variable for different parts of the HCV phylogeny (*Simmonds et al., 2005*). The greatly overlapping mPTP and ABGD clusters with the HCV genotypes shows that the classification, and, consequently, the identification, of the genotypes can be easily automated utilizing objective, transparent, and unifying approaches. Embracing such alternatives can be crucial for viruses like HCV, taking into account that the correct identification of the HCV genotypes could be of clinical importance in providing the appropriate medical treatment (*Strader et al., 2004*; *Ge et al., 2009*).

### HBV clustering

In the case of HBV, the mPTP clustering is almost identical to the current classification (*Norder et al., 2004*; *Kramvis & Kew, 2007*) of the virus that comprises eight genotypes, except for subgenotype C4 which formed a new cluster (Fig. 3, Figs. S3 and S4 and Appendix S1). This is in line with the greater genetic divergence of C4 compared to the other subgenotypes due to its ancient origin in native populations in Oceania (*Paraskevis et al., 2013*). However, the split of C4 from its sister cluster (genotype C) is not supported by the MCMC sampling, potentially reflecting the lack of adequate sampling. On the other hand, the number of clusters identified by ABGD varied from 1 to 85 under different thresholds of minimum intraspecific divergence, while the delimitation for a threshold of 1.29% exactly matched the eight genotypes of the HBV classification (Figs. 2 and 3). Both ABGD and mPTP identified seven of the genotypes (A–F) as distinct genetic clusters. The only difference was that mPTP split genotype C into two distinct clusters (Fig. 3), i.e., subgenotype C4 was recovered as a distinct cluster from the remaining seven subgenotype.

## CONCLUSIONS

The application of mPTP to the HCV and HBV data sets shows that automated viral species delimitation using phylogeny-aware methods yields clusters that largely agree with the current standard taxonomy. The additional clusters identified for HCV by mPTP is not surprising as they have been previously considered divergent sub-clusters within the genotypes 3 and 6. Analogously, for HBV, mPTP yielded almost identical results to the current nomenclature system with the exception of a single sub-genotype, C4, that was previously mentioned to be more genetically divergent within genotype C (*Paraskevis et al., 2013*). In both cases, these new clusters indicate the potential need for taxonomic revision. However, given the wide use of the current nomenclature in the medical field, and the lack of other sources of information such as recombination, particularly for HBV, and, response to treatment, we wouldn't suggest taxonomic changes at present. Regarding distance methods, the example of HCV and HBV, shows that meaningful parameter values for distance-based methods may differ substantially among datasets, and, therefore, establishing global thresholds is impossible. On the contrary, mPTP can be seamlessly applied to taxa of substantially different life histories (e.g., variable population sizes, evolution rates), as it does not require any input parameters except a phylogeny. Overall, the ease-of-use of mPTP in conjunction with its computational efficiency on phylogenies with hundreds of samples render it a useful tool for viral biodiversity estimates, initial classification of understudied taxa, and accelerating the viral species identification process.

## ACKNOWLEDGEMENTS

We would like to thank the editor and the reviewers for their valuable comments and suggestions that helped improve and clarify the manuscript.

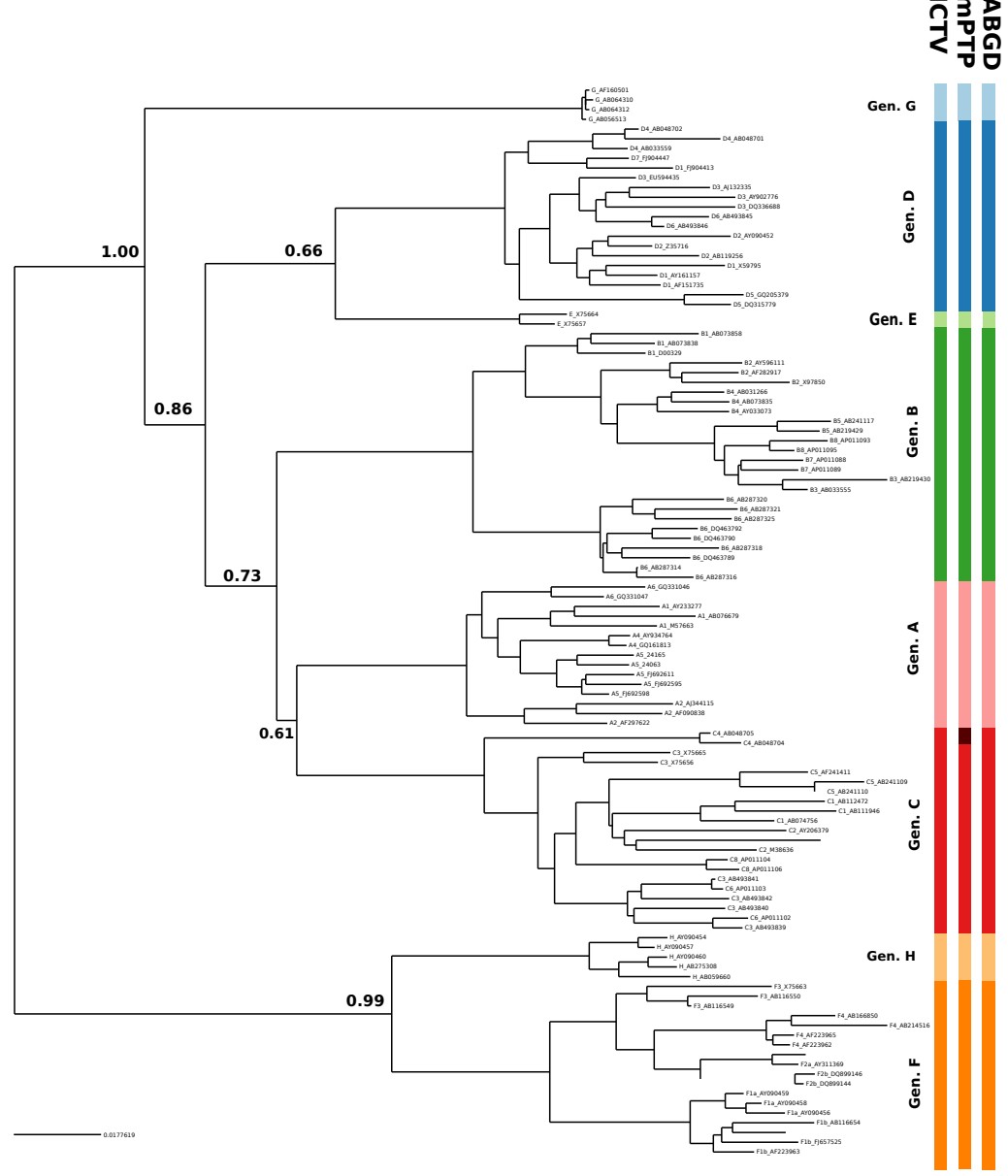

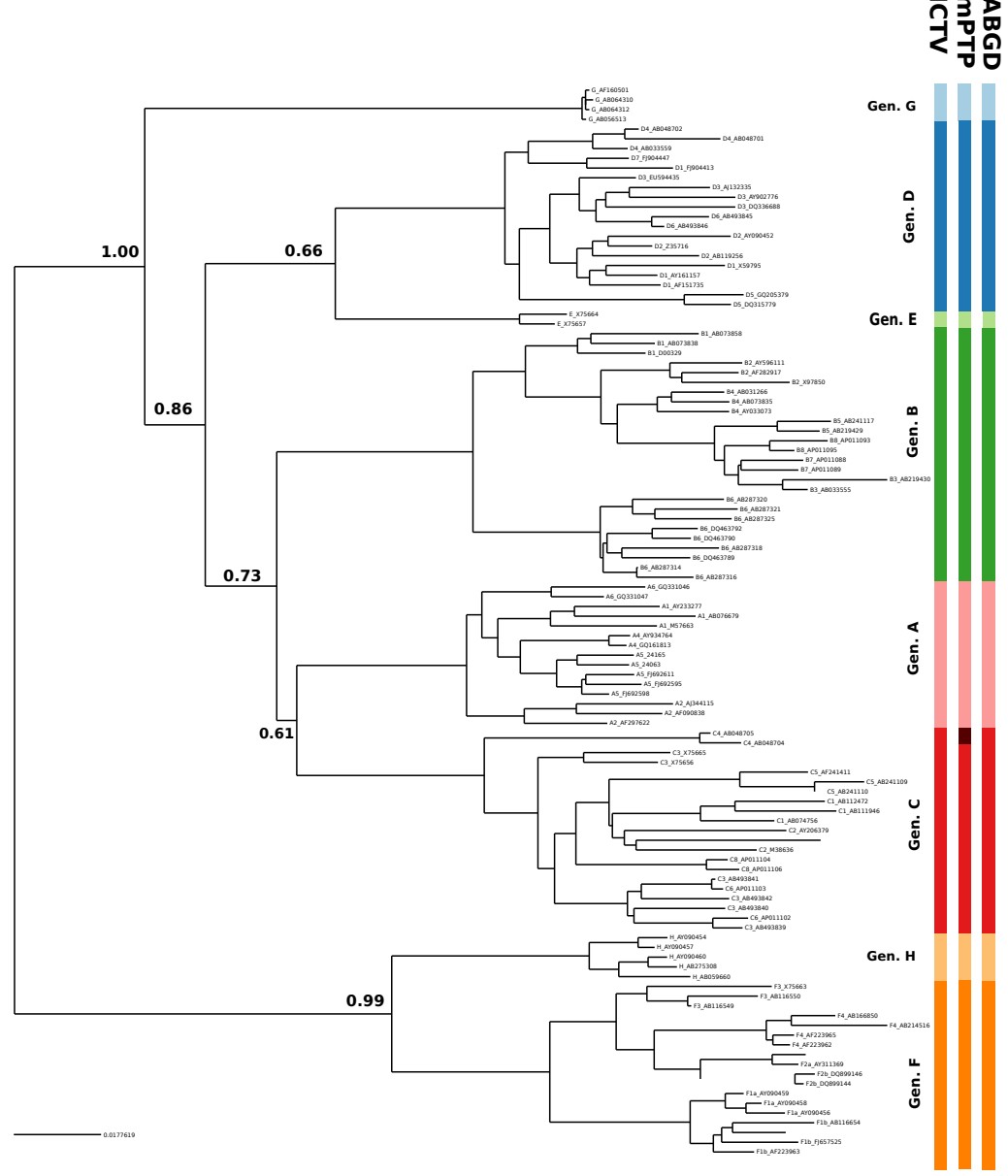

**Figure 3 Clustering of the HBV samples into genotypes; the first colored bar corresponds to the genotypes currently acknowledged by ICTV, the second to the mPTP ML clustering and the third to the ABGD clustering ($p = 0.0129$, $X = 0.5$).** The numbers indicate the support for a particular node being a speciation node obtained by the MCMC sampling under the mPTP model (support $<0.5$ not shown, but see Fig. S4). The phylogenetic relationships were inferred using RAxML under the GTR+Γ model.

### Funding

This work was supported by the Klaus Tschira Foundation. The funders had no role in study design, data collection and analysis, decision to publish, or preparation of the manuscript.

### Grant Disclosures

The following grant information was disclosed by the authors:
Klaus Tschira Foundation.

### Competing Interests

The authors declare there are no competing interests.

### Author Contributions

- Dora Serdari analyzed the data, prepared figures and/or tables, authored or reviewed drafts of the paper, approved the final draft.
- Evangelia-Georgia Kostaki analyzed the data, authored or reviewed drafts of the paper, approved the final draft.
- Dimitrios Paraskevis authored or reviewed drafts of the paper, approved the final draft.
- Alexandros Stamatakis conceived and designed the experiments, authored or reviewed drafts of the paper, approved the final draft.
- Paschalia Kapli conceived and designed the experiments, analyzed the data, prepared figures and/or tables, authored or reviewed drafts of the paper, approved the final draft.

### Data Availability

The sequence data used in this study were downloaded from NCBI. The relevant Accession Numbers are available in the Supplementary Appendix. The resulting data from the phylogenetic and clustering analyses are available in the GitHub repository: https://github.com/Pas-Kapli/peerj-mptp-HBV-HCV-data.

### Supplemental Information

Supplemental information for this article can be found online at http://dx.doi.org/10.7717/peerj.7754#supplemental-information.

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
