# Peer review of "Automated, phylogeny-based genotype delimitation of the Hepatitis Viruses HBV and HCV"

_PeerJ, doi:10.7717/peerj.7754_

## Round 0.1 · original submission · Minor Revisions

Please regard that among other corrections, you need provide the cutoff and support values of the subtrees and more specific branches inside the phylogenetic trees in Figures 1 and 2.

·

Basic reporting

Clear and professional English language used throughout. Data and images were checked and seem right. Intro & background are suitable. Raw data supplied.
Some minor changes are described below.
- Add a footnote with a short description for the supplementary figures.
- Review the bibliography format, for example on line 507 does not have the title of the paper (507 Thézé, J., Lopez-Vaamonde, C., Cory, J. S., & Herniou, E. A. (2018). Viruses, 10(7), 366).

Experimental design

No comment

Validity of the findings

The data is robust. The conclusions are well established, linked to the original research question and limited to the results of support. However, a small paragraph about the differences between current ICTV classification and mPTP method could be added in the discussion of the results. Should the new clusters they detected be added? Or should they not be taken into account? In case the last option is correct, why then does mPTP separate them as different clusters? It's a mistake?

Comments for the author

The manuscript entitled " Automated, phylogeny-based genotype delimitation of the Hepatitis Viruses HBV and HCV” by Serdari and collaborators, evaluates the use of computational methods for molecular species delimitation in hepatitis virus classification. They use two correctly selected data sets of HBV and HCV and two different methods (mPTP and ABGD), then comparing the results obtained with the current ICTV classification.
In my view, the paper presents a good development and analysis of the results obtained. The manuscript is interesting and provides important information that increases knowledge about the use of the multi-rate Poisson Tree Processes and the classification of hepatitis viruses. So it would be useful for virologists and bioinformatics researchers. Therefore, it could be accepted. Some minor changes are described below.
- Add in the discussion about the discrepancies between the current ICTV classification and those proposed by the mPTP method, a small paragraph mentioning what may be due. Should the new clusters they detected be added? Or should they not be taken into account? In case the last option is correct, why then does mPTP separate them as different clusters? It's a mistake?
- Add a footnote with a short description for the supplementary figures
- Review the bibliography format, for example on line 507 does not have the title of the paper (507 Thézé, J., Lopez-Vaamonde, C., Cory, J. S., & Herniou, E. A. (2018). Viruses, 10(7), 366).

Reviewer 2 ·

Basic reporting

No comment.

Experimental design

No comment.

Validity of the findings

I strongly suggest the authors clarify the cutoff value used for showing/not showing support values in Figures 1 and 2.
Similarly, it would be interesting to know the support values for the sub-trees corresponding to all the genotypes shown in Figures 1 and 2 (irrespective of cutoff value used).

·

Basic reporting

The study assessed whether the multi-rate Poisson Tree Processes (mPTP) algorithm can be used in the classification of viruses according to ICTV.
Authors used sequences of hepatitis B and C viruses (HBV and HCV, respectively) to test the algorithm. The distance-based “Automatic Barcode Gap Discovery” tool (ABGD) was also used in the analysis to compare with mPTP.
Of note, analysis using mPTP is based on the input of a phylogenetic tree, whereas for ABGD the user must define other specific parameters. Results showed that HBV and HCV classification using mPTP corresponds well to the ICTV classification,
However, minor errors were found along the text, which should be correct before the article is published. Corrections were suggested electronically in the pdf file.

Experimental design

The study is well designed.

Validity of the findings

Results are robust and well explained.

Comments for the author

The study assessed whether the multi-rate Poisson Tree Processes (mPTP) algorithm can be used in the classification of viruses according to ICTV.
Authors used sequences of hepatitis B and C viruses (HBV and HCV, respectively) to test the algorithm. The distance-based “Automatic Barcode Gap Discovery” tool (ABGD) was also used in the analysis to compare with mPTP.
Of note, analysis using mPTP is based on the input of a phylogenetic tree, whereas for ABGD the user must define other specific parameters. Results showed that HBV and HCV classification using mPTP corresponds well to the ICTV classification,
The study is well designed, results are robust and well shown and the article is well written. However, minor errors were found along the text, which should be correct before the article is published.
Corrections were suggested electronically in the pdf file.

---

## Round 0.2 · Minor Revisions

Only minor remaining corrections are needed,

·

Basic reporting

no comment

Experimental design

no comment

Validity of the findings

no comment

Comments for the author

I believe that the authors have improved the work according to the indications of the reviewers. Likewise, the responses by the authors contemplate the data requested and the doubts raised.
Therefore, I consider that the work "Automated, phylogeny based genotype delimitation of Hepatitis Viruses HBV and HCV" can be accepted for publication in PeerJ.

·

Basic reporting

The authors made several corrections along the manuscript according to the reviewers comments and suggestions. However, I believe that they did not understand the corrections and suggestions that I made on the pdf file, therefore I re-corrected in the Word document file, using track changes.
In my opinion, after these changes are made, the manuscript is suitable for publication.

Experimental design

Good

Validity of the findings

Good

Comments for the author

The authors made several corrections along the manuscript according to the reviewers comments and suggestions. However, I believe that they did not understand the corrections and suggestions that I made on the pdf file, therefore I re-corrected in the Word document file, using track changes.
In my opinion, after these changes are made, the manuscript is suitable for publication.

---

## Round 0.3 · accepted · Accept

Your manuscript describes a valuable contribution for those working in routine characterization of HBV and HCV samples.